# Association between Cervical Cancer and Dietary Patterns in Colombia

**DOI:** 10.3390/nu15234889

**Published:** 2023-11-23

**Authors:** Luz Adriana Meneses-Urrea, Manuel Vaquero-Abellán, Dolly Villegas Arenas, Narly Benachi Sandoval, Mauricio Hernández-Carrillo, Guillermo Molina-Recio

**Affiliations:** 1Research Group “Health Care (Recognized by Colciencias)”, Universidad Santiago de Cali, Cali 760001, Colombia; luz.meneses00@usc.edu.co (L.A.M.-U.); dollyvillegas8@gmail.com (D.V.A.); nbenachi@clinic.cat (N.B.S.); 2Department of Nursing, Universidad Santiago de Cali, Cali 760001, Colombia; 3IMIBIC GC12 Clinical and Epidemiological Research in Primary Care (GICEAP), 14014 Córdoba, Spain; 4Department of Nursing, Pharmacology and Physiotherapy, University of Córdoba, 14014 Córdoba, Spain; gmrsurf75@gmail.com; 5CAP Casanova, Consorci d’Atenció Primària de Salut Barcelona Esquerra, 08036 Barcelona, Spain; 6Health Faculty, Universidad del Valle, Cali 760001, Colombia; mauriciohc@gmail.com; 7Health Faculty, Escuela Nacional del Deporte, Cali 760001, Colombia; 8Lifestyles, Innovation and Health (GA-16), Maimonides Biomedical Research Institute of Córdoba (IMIBIC), 14014 Córdoba, Spain

**Keywords:** food consumption, cervix uterine, neoplasm, diet, risk factors

## Abstract

Cervical cancer is a global public health problem. It is the second leading cause of death among women of childbearing age worldwide. Several factors, including diet, have been shown to influence the risk of persistent HPV infection and tumor progression. This paper determines the relationship between dietary patterns and cervical cancer. It is an ecological study of multiple groups, based on two national sources: the High-Cost Account and the National Survey of Nutritional Situation of Colombia of 2015. The population consisted of 3472 women aged 35 to 64. The incidence of cervical cancer was used as the dependent variable while the independent variables included food consumption according to established patterns, area of residence, age, physical activity, and BMI, among other variables. The statistical analysis performed through associations between variables was evaluated by multiple linear regression using R2. 38.9% of the evaluated population belonged to the first quartile of wealth, and 76.5% resided in the municipal capital. The incidence of cervical cancer in Colombia was associated with being affiliated to a state-subsidized health regime and having diabetes mellitus. A conservative eating pattern, as well as belonging to a rural area, are evidenced as protective factors. These results invite the need to encourage public policies and promote healthy lifestyles.

## 1. Introduction

Cervical cancer is a global public health problem, particularly in low- and middle-income countries (LMICs) [1]. It is the second most frequent and the second leading cause of death in women of reproductive age in the world [2]. In 2020, an incidence of 604,000 and 342,000 deaths was estimated [3]. According to GLOBOCAN data for 2020, Latin America and the Caribbean ranked second worldwide in incidence and mortality, with 59,439 new cases and 31,582 deaths [4]. It has been suggested that the incidence in these LMICs may be due to several factors, including lifestyle [5].

Human papillomavirus (HPV) is one of the causative agents of cervical cancer and precursor lesions [6]. Several causes have been shown to affect the risk of persistent HPV infection and the progression of neoplasms, including diet [7,8]. HPV survives in a cellular oxidative environment and has been shown to be more likely to evolve into neoplasms by increasing oxidative stress and DNA damage [9]. For these reasons, dietary intake of antioxidants appears to exert a protective effect on HPV infection and persistence [7,8]. According to the results of the EPIC (European Prospective Investigation into Cancer and Nutrition) study, there is a significant inverse association between invasive squamous cervical cancer and daily increases in fruit intake [10].

Regarding dietary patterns, a recent study has established that adhering to the Mediterranean diet reduces the risk of cervical cancer by 60%, while following a Western diet represents a risk factor for its appearance [11]. It is estimated that changes in dietary habits can contribute to preventing the onset of cancer by 30% to 40% [12].

For these reasons, this research was proposed to determine the relationship between dietary patterns and cervical cancer, considering other sociodemographic aspects of different regions in Colombia.

## 2. Methodology

A multi-group ecological study was conducted based on two national sources, HIGIA (High-Cost Account), and ENSIN 2015 (National Survey of Nutritional Situation of Colombia of 2015). The population consisted of 3472 women aged 35 to 64, distributed across six regions of Colombia: Atlántica, Central, Oriental, Pacífica, Orinoquía, and the Capital District of Bogotá.

The incidence of cervical cancer was used as the outcome variable, taken from the High-Cost Account. Independent variables were obtained from the National Survey of the Nutritional Situation (ENSIN) 2015, including: area of residence, affiliation regime, quartile of wealth, educational level, age ranges, compliance with the recommendations of physical activity, compliance with 30 min of physical exercise per day for at least 5 days, following the recommendations of vigorous physical activity, and BMI according to World Health Organization (WHO) categories [13]. In addition, information was collected on the suffering of hypertension, diabetes mellitus, following a special diet, and following a vegetarian diet. These variables were dichotomized. 

The area of residence was structured as a categorical variable, including municipal seat (defined as the urban perimeter limited by agreements of the Municipal Council, where the administrative headquarters of the municipality is located), populated center (refers to the concentration of at least twenty contiguous houses, neighboring or attached to each other, located in the rural area of a municipality or a departmental corregimiento), and rural center, defined by dispersed dwellings with disaggregated farms [14].

The independent variable, consumption of dietary patterns, was taken from the work of Meneses et al., who established four dietary patterns: traditional, industrialized, conservative, and beverages/grilled foods. The traditional pattern was made up of the dairy group, potato/legume groups, cereal group, fried group, coffee/panela, and meat-sausages. The industrialized pattern was characterized by the consumption of soft drinks, fast foods, sweets, and package foods. The conservative pattern referred to individuals who based their diet on the group of vegetables, fruits, light foods, supplements, and whole foods. Finally, the beverages/grilled foods pattern was made up of the group of energy drinks, alcohol, and grilled foods [15]. To introduce the patterns into the statistical models, it was necessary to generate a new dichotomous dependent variable for each one, through the binomial distribution, as follows: those presenting some degree of consumption of the corresponding pattern were categorized as 1, while those who did not report any level of consumption of the pattern were categorized as 0.

The statistical analysis was performed through the association of contextual variables and was evaluated by multiple linear regression using R2 and the principle of statistical parsimony for multiple linear regression as a goodness of fit measure, which considers the minimum sum of squares [16]. 15 models were performed until the best one was obtained, representing the dietary patterns and accounting for possible confounding effects of the contextual variables on the incidence of cervical cancer. The statistical modeling used a stepwise method, with a criterion of eliminating probability values above 0.10.

### Ethical Considerations 

The study was endorsed by the Institutional Ethics Committee of the Universidad Santiago de Cali, Approval Act No. 11 of 29 May 2020. The database used was authorized by the Colombian Institute of Family Welfare (ICBF) to conduct the research, and it involved working with available anonymized data, with the consent (verbal or signed) of the participants in these national surveys to the interviewers of the national health system.

## 3. Results

Information was collected from a sample of 3472 women within the age range of 35 to 64. Through the evaluation of sociodemographic variables, relevant findings are highlighted, including the distribution of wealth, educational level, age, geographical location, and health insurance regime. The results indicate that 38.9% of the evaluated population belonged to the first quartile of wealth (Q1), and the region with the highest percentage in the first quartile of wealth (Q1) was Amazonia-Orinoquia, with 53.5%. 71.4% of respondents had an educational level up to high school, and 53% were in the age range of 35 to 49, occurring most frequently in the Atlantic region (64.7%). In addition, 76.5% resided in municipal capitals, with the Pacific region having the most reported living in rural areas at 32%. 58.1% of the participants belonged to the subsidized health care regime or were unaffiliated. It is important to note that this regime is used by the Colombian State to provide access to health services through granted subsidies for those without the ability to pay. The region with the highest percentage under this regime was the Amazonía-Orinoquia region, at 69.5% (Table 1).

Likewise, when considering the additional variables evaluated in the study, it was found that 12.6% of the population complied with the recommendations of physical activity, with 1.6% engaging with physical activity for at least 30 min, five times a week, and only 0.7% participating in vigorous physical activity. On the other hand, 100% of the sample did not follow a vegetarian diet or any kind of special diet. In relation to the prevalence of chronic diseases, 48.8% reported having hypertension, 2.7% had diabetes, 38.1% were overweight, and 76.3% were obese. Additional data structured by region are available in Table 2.

In relation to dietary patterns, 94.6% of the population had some consumption of pattern 1 (traditional), 92.9% of pattern 3 (conservative), 73.9% of pattern 2 (industrialized), and 57.7% had an intake linked to pattern 4 (food, beverages/grill). The traditional and conservative patterns stood out in the Pacific region, while the industrialized pattern was of higher consumption in the Atlantic region and the beverage/grill food pattern was reported more frequently in the Eastern region. The Department of Tolima, despite belonging to this region, reported no consumption. The Bogotá region was characterized by pattern 2 (industrialized) and the Amazonia-Orinoquia region, compared to the other regions, had the highest consumption of beverage/grilled foods (see Figure 1 and Table 3).

Regarding the incidence of cervical cancer in people over 35 years of age in 2020, the highest rate was found in the Amazonia-Orinoquia region, with 23.8 cases per 100,000 habitants. This region is the one that reported the highest percentage of the population in Q1 of wealth, affiliation to the subsidized health regime, declarations of diabetes, and consumption of beverage foods/grill. The lowest incidence rate was reported in the Eastern region, with 12.8 per 100,000 hb. This region had the lowest prevalence of obesity (see Figure 2).

Modelling was performed using an initial saturated model that contained all available explanatory variables [16]. After the variables in the multivariate model were adjusted, 15 models were generated. 68% of the incidence of cervical cancer can be explained by the explanatory variables of the multiple linear model (Table 4), showing a positive association with certain factors, including belonging to the subsidized regime or not being affiliated (*p* = 0.002), and having diabetes (*p* = 0.07). In addition, a negative association was found with the conservative dietary pattern (*p* = 0.013) and residing in a populated or dispersed rural center (*p* = 0.003), (Table 5).

## 4. Discussion

An ecological study was conducted using secondary data from the database of the National Survey of the Nutritional Situation (ENSIN) 2015 and the 2020 incidence rate of cervical cancer for people over 35 years of age, using the High-Cost Account as a source of information.

The incidence rate of cervical cancer has risen worldwide, especially in countries with a high Human Development Index (HDI). Colombia is considered a country with a high HDI [17]. This rate is influenced by aging, population growth, and changes in cancer risk factors, some of which are related to socioeconomic development [18] 

The findings of this research showed associations between the incidence rate and variables including the area of residence, populated or rural center, subsidized health regime, having diabetes, and the consumption of the conservative pattern. The results of our study found no association with the quartile of wealth, despite the fact that 62.4% of the population was in quartiles 1 and 2 of wealth, which may be due to early detection and efficient treatment [19]. Several studies have reported that the probability of developing cancer is higher in socially vulnerable populations and that the risk of getting sick and dying is associated with lower levels of income, education, and higher degrees of social discrimination [20,21].

The lack of adequate health insurance coverage is a challenge in communities of poverty and is associated with reduced access to care, prevention, and diagnosis. Patel et al. conducted a study in California and reported that patients with *Medicaid*, insurance granted by the state, or those without any medical insurance had a higher incidence of cervical cancer [22], which coincides with the results of our research. Another study in the United States showed that *Medicaid* patients had a 38% increase in mortality (HR = 1.38; CI 95% 1.34–1.43), while uninsured patients had a 32% increase (HR = 1.32; CI 95%, 1.26–1.38), which may be associated with lower quality of care and reduced adherence to medical protocols in public hospitals [23]. On the other hand, the report on the situation of cancer in Colombia in 2021 reported that the incidence of cervical cancer was similar for both the contributory and subsidized regimes [24].

Regarding the area of residence, our research found that living in a populated/rural center decreases the incidence of cervical cancer, which coincides with the findings of Hall et al., who conducted a study in Florida, United States, using Cancer Data System data during the years 2014–2018. They sought to determine age-adjusted incidence and mortality for 22 types of cancer, including cervical cancer. These authors found that the incidence was significantly lower in rural areas than in urban areas [25]. In the same way, the study of Fantin et al., with cancer registries from 2011 to 2015 in Costa Rica, showed that cervical cancer was higher in urban areas [26]. In any case, it should be noted that the results of lower incidence in rural areas may be due to underreporting or poor accessibility to health services in these areas.

According to WHO recommendations, physical activity confers several benefits, including reducing mortality from any cause, cardiovascular diseases, hypertension, specific cancers, and type 2 diabetes, in addition to contributing to mental health. For these reasons, the WHO emphasizes that adults should perform moderate activity for two or more days a week, with a duration of 150–300 min of moderate to intense activity, or 75–150 min of vigorous activity to obtain these benefits [27]. In our research research, we found that only 8.5% meet these recommendations for physical activity, 0.8% meet the recommendations for 30 min of physical activity five times a week and 0.5% meet the recommendations for vigorous activity. These figures reflect a significant decrease, taking into account that in Colombia in 2010, the prevalence of moderate physical activity per week was 53.5% [28]. In high-income countries, the prevalence of physical inactivity was double (36.8%, 35–38) that of LMICs (16.2%, 14.2–17.9) [29]. Despite this, our study did not find an association between cervical cancer and physical activity, possibly due to the low percentage of the population that performs it regularly, which may have been decisive in finding a statistical relationship. Contrary to these findings, Lee et al. in their study in Korea, using cancer registries from 2006 to 2012 in women with intracervical neoplasia and cancer, showed that the risk of grade 2 and 3 intracervical neoplasia was inversely associated with physical activity [30].

With regard to BMI, our results showed no association with cervical cancer. Wise et al. and Emma et al. found an association between a BMI greater than 30 kg/m^2^ and endometrial cancer [31,32]. Similarly, high BMI is considered one of the risk factors for all types of cancer, according to the Global Burden of Cancer [33]. The difference in results may be due to the ecological, pooled, and cross-sectional data used in this study, which has been able to reduce the measurement of the effect of obesity on the incidence of cervical cancer. However, the research discussed above [33,34] corresponds to longitudinal and individual studies, where such types of biases are usually controlled.

In relation to the disease declared in our study, diabetes was associated with an increased incidence of cervical cancer, which may be related to hyperglycemia and chronic inflammation caused by this pathology [34], in addition to the imbalance in the proteasome system among these patients, taking into account that the development of cancer requires high protein turnover and involves this system [35]. In a study by Garduño et al. in Mexico involving women with cervical cancer, an association with diabetes was found [36]. These results also coincide with those of Yang et al., who conducted a study in Taiwan with adults over 35 years diagnosed with several types of cancers, including cervical cancer, and found a similar association (adjusted RR 13.4; CI 95% 2.70–66.3) [37].

Regarding the incidence of cervical cancer, the highest rate in 2020 was found in the Amazon-Orinoquia region, in which the highest proportion of individuals was found in Q1, with affiliations to the subsidized health regime, a higher percentage of diabetes in relation to other regions, and more frequently reported consumption of beverage/grill foods. Studies have reported the presence of polycyclic aromatic hydrocarbons (PAHs) and heteroaromatic amines (HAAs) in meat that has been exposed to high temperatures during grilling [38,39]. One of the PAHs, benzo (a) pyrene (BaP), is known as a carcinogen, capable of modulating cellular processes including differentiation, proliferation, immune response, cancer promotion, and apoptosis [40,41].

On the other hand, according to the National Quality of Life Survey in 2018, in the Orinoquia-Amazon region, 69.9% of people aged 2 years and over reported consuming sugary drinks, a figure that exceeds the national record of 68.4% [42]. 

A diet high in sugar usually leads to weight gain and metabolic parameters associated with obesity, insulin resistance, the bioactivity of steroid hormones, oxidative stress, inflammation, and ultimately, the development and progression of cancer [43]. According to Choi et al., the consumption of sugary drinks is associated with type II endometrial cancer [44]. Therefore, the incidence of cervical cancer in this region may be associated with the consumption of sugary drinks.

In terms of dietary patterns, it was found that the conservative pattern is related to a lower incidence of cervical cancer. These results are consistent with Marziyeh et al., who showed that a high consumption of fruits and vegetables can reduce the risk of cervical cancer [45]. In addition, Barchitta et al., concluded that the risk of cervical cancer was lower in patients with high scores in the healthy diet pattern [8], while Mogge et al. found that patients with cervical cancer had a lower intake of fruits and vegetables [46]. In addition, Hwang et al. found that fruit and vegetable consumption reduces the risk of cervical cancer (OR = 2.84, 95% CI 1.26 to 6.42, *p* = 0.06 for vegetables; OR = 2.93, 95% CI 1.25 to 6.87, *p* = 0.01 for fruits) [47]. All of the above is possible because of the antioxidants that fruits and vegetables have, such as vitamin C and a- and b-carotene, which limit the damage caused by free radicals [48].

### Limitations

Studies based on nutritional patterns rather than the measurement of dietary intakes do not allow us to know what type of foods or nutritional values could be more closely related with the development of this type of cancer. In any case, they are frequently used tools in ecological designs based on health surveys, as they represent food and nutrient consumption from a more comprehensive and community-based view on diet as a factor of exposure to a health event [49,50]. For these reasons, there is a need for longitudinal studies to establish a more precise association between dietary intakes and the development of cervical cancer. Another notable limitation is the lack of data on medical and gyneco-obstetrical factors, which were not collected in the health survey used for this secondary analysis. These factors could also be related to the development of this type of cancer. On the other hand, because it is a secondary source, the participants’ responses may have recall bias. It should be noted that the sample size of the study decreases the probability of incurring type II errors. 

## 5. Conclusions

It is concluded that, at the ecological level, the increase in the incidence of cervical cancer in Colombia was associated with affiliation to a health regime subsidized by the state and comorbidity with diabetes mellitus. In addition, the conservative dietary pattern, consisting of fruits and vegetables, as well as belonging to a rural area, were evidenced as protective variables. These results invite us to encourage public health policies that reduce population inequity and to make educational efforts for the promotion of healthy lifestyles to prevent cervical cancer.

## Figures and Tables

**Figure 1 nutrients-15-04889-f001:**
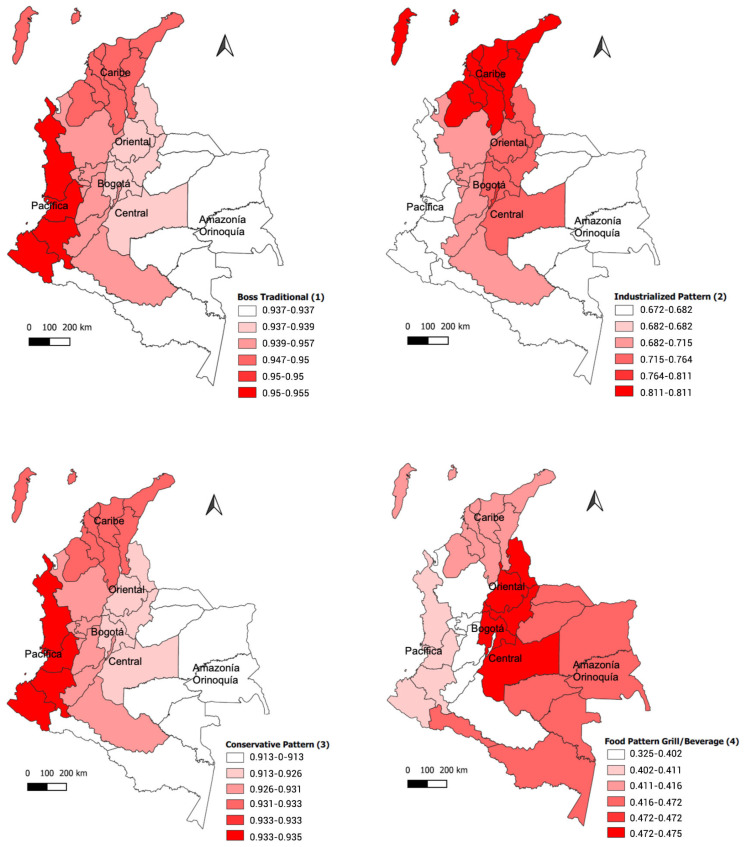
Dietary pattern by region.

**Figure 2 nutrients-15-04889-f002:**
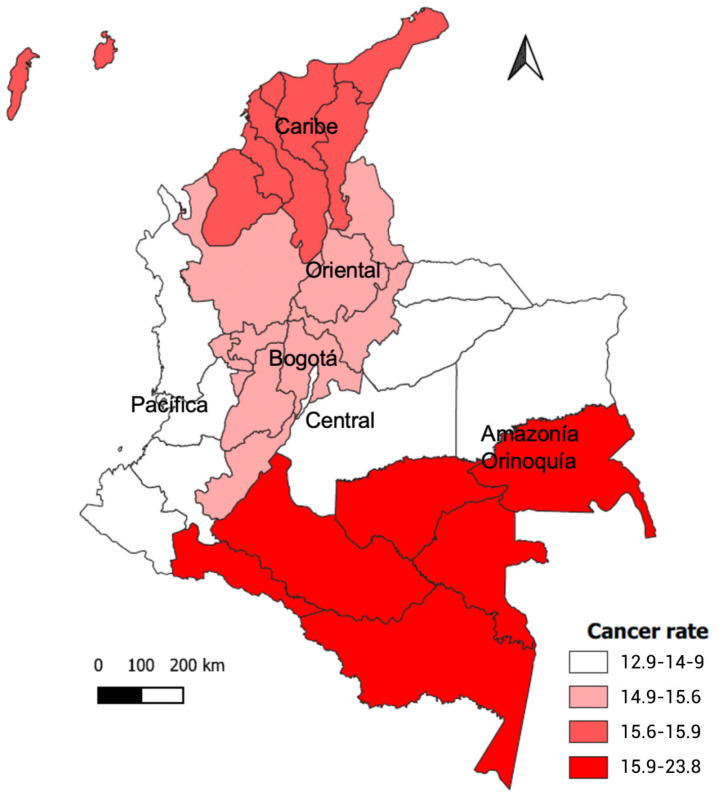
Incidence of cervical cancer by region.

**Table 1 nutrients-15-04889-t001:** Sociodemographic characteristics.

Variable	Category	Region													
		Atlántico		Oriental		Amazonia-Orinoquía		Bogotá		Central		Pacífica		Total	
		n = 762	%	n = 609	%	n = 462	%	n = 243	%	n = 887	%	n = 509	%	n	%
Quartile of wealth	First quartile	363	47.6	208	34.2	247	53.5	6	2.5	286	32.2	241	47.3	1351	38.9
	Second quartile or more	399	52.4	401	65.8	215	46.5	237	98	601	19.8	268	52.7	2121	64.8
Educational level	Less than primary (0–4 years)	225	29.5	163	26.8	134	29	29	11.9	238	26.8	184	36.1	973	28
	Primary/Secondary complete	534	70.1	443	72.7	324	70.1	214	88.1	641	72.3	322	63.3	2478	71.4
Age ranges	35–39	198	26	118	19.4	90	19.5	49	20.2	163	18.4	86	16.9	704	20.3
	40–44	142	18.6	93	15.3	74	16	38	15.6	140	15.8	64	12.6	551	15.9
	45–49	153	20.1	91	14.9	73	15.8	42	17.3	169	19.1	68	13.4	596	17.2
	50–54	93	12.2	132	21.7	82	17.7	44	18.1	162	18.3	116	22.8	629	18.1
	55–59	97	12.7	92	15.1	86	18.6	41	16.9	132	14.9	87	17.1	535	15.4
	60–64	79	10.4	83	13.6	57	12.3	29	11.9	121	13.6	88	17.3	457	13.2
Area	City	585	76.8	409	67.2	441	95.5	243	100	663	74.7	315	61.9	2656	76.5
	Village/Rural	177	23.2	200	32.8	21	4.5	0	0	224	25.3	194	38.1	816	23.5
Health affiliation	Subsidized/nonaffiliated	500	65.6	320	52.5	321	69.5	71	29.2	478	53.9	326	64.0	2016	58.1
	Contributory	262	34.4	289	47.5	141	30.5	172	70.8	409	46.1	183	36	1456	41.9

**Table 2 nutrients-15-04889-t002:** Lifestyle and health status.

Variable	Category	Region						
		Atlántico n = 762	Oriental n = 609	Amazonia-Orinoquía n = 462	Bogotá n = 243	Central n = 887	Pacífica n = 509	Total
		n(%)	n(%)	n(%)	n(%)	n(%)	n(%)	n(%)
Meets physical activity recommendations	Complies	65(8.5)	86(14.1)	62(13.4)	49(20.2)	117(13.2)	59(11.6)	438(12.6)
	Does not comply	697(91.5)	523(85.9)	400(86.6)	194(79.8)	770(86.8)	450(88.4)	3034(87.4)
Physical activity 30 min per day (at least 5 days)	Complies	6(0.8)	15(2.5)	6(1.3)	6(2.5)	9(1.0)	12(2.4)	54(1.6)
	Does not comply	756(99.2)	594(97.5)	456(98.7)	237(97.5)	878(99.0)	497(97.6)	3418(98.4)
Meets recommendations for vigor	Complies	4(0.5)	4(0.7)	7(1.5)	2(0.8)	6(0.7)	3(0.6)	26(0.7)
	Does not comply	758(99.5)	605(99.3)	455(98.5)	241(99.2)	881(99.3)	506(99.4)	3446(99.3)
Overweight classification	Yes	284(37.3)	237(0.4)	167(36.1)	99(0.4)	331(37.3)	206(0.4)	1324(38.1)
	No	478(62.7)	372(0.6)	295(63.9)	144(0.6)	556(62.7)	303(59.5)	2148(61.9)
Obesity classification	Yes	222(29.1)	146(0.2)	153(33.1)	47(19.3)	253(28.5)	150(29.5)	971(28)
	No	540(70.9)	463(0.8)	456(98.7)	196(80.7)	634(71.5)	359(0.7)	2648(76.3)
Overweight/Obesity	No risk	174(25)	163(29.4)	104(24.2)	73(32.9)	224(25.3)	107(22.8)	845(24.3)
	At risk	523(75)	391(70.6)	326(75.8)	149(67.1)	592(72.5)	362(77.2)	2343(67.5)
Thinness	No risk	748(98.2)	608(99.8)	459(99.4)	243(100)	884(99.7)	506(99.4)	3448(99.3)
	At risk	12(1.6)	1(0.2)	3(0.6)	0(0)	3(0.3)	3(0.6)	22(0.7)
Hypertension	No	449(58.9)	255(41.9)	267(57.8)	94(38.7)	451 (50.8)	262(51.5)	1778 (51.2)
	Yes	313(41.1)	354(58.1)	195(42.2)	149(61.3)	436(49.2)	247(48.5)	1694(48.8)
Diabetes	No	746(97.9)	594(97.5)	437(94.6)	236(97.1)	870(98.1)	494(97.1)	3377(97.3)
	Yes	16(2.1)	15(2.5)	25(5.4)	7(2.9)	17(1.9)	15(2.9)	95(2.7)
You follow a special diet	Yes	0(0)	0(0)	0(0)	0(0)	0(0)	0(0)	0(0)
	No	762(100)	609(100)	462(100)	243(100)	887(100)	509(100)	3472(100)
You are a vegetarian	Yes	0(0)	0(0)	0(0)	0(0)	0(0)	0(0)	0(0)
	No	762(100)	609(100)	462(100)	243(100)	887(100)	509(100)	3472(100)

**Table 3 nutrients-15-04889-t003:** Eating patterns.

Variable	Category	Region													
		Atlántico		Oriental		Amazonia-Orinoquía		Bogotá		Central		Pacífica		Total	
		n = 762	%	n = 609	%	n = 462	%	n = 243	%	n = 887	%	n = 509	%	n	%
Pattern Consumption 1	Yes	724	95	572	93.9	433	93.7	230	94.7	840	94.7	486	95.5	3285	94.6
	No	38	5.0	37	6.1	29	6.3	13	5.3	47	5.3	23	4.5	187	5.4
Pattern Consumption 2	Yes	618	81.1	465	76.4	315	68.2	191	78.6	634	71.5	342	67.2	2565	73.9
	No	144	18.9	144	23.6	147	31.8	52	21.4	253	28.5	167	32.8	907	26.1
Pattern Consumption 3	Yes	711	93.3	564	92.6	422	91.3	226	93	826	93.1	476	93.5	3225	92.9
	No	51	6.7	45	7.4	40	8.7	17	7	61	6.9	33	6.5	247	7.1
Pattern Consumption 4	Yes	317	41.6	289	47.5	218	47.2	79	32.5	357	40.2	209	41.1	1469	42.3
	No	445	58.4	320	52.5	244	52.8	164	67.5	530	59.8	300	58.9	2003	57.7

**Table 4 nutrients-15-04889-t004:** Comparison of ecological multiple linear regression models for cervical cancer incidence, adjusting for dietary patterns and other variables. Regions of Colombia. Year 2020.

Model	# Var	# Sig Var	R	R Square *	Adjusted R-Square	Standard Error of Estimation
1	19	0	0.889	0.78972641	0.54002652	3.35422758
2	18	0	0.889	0.78972435	0.56707953	3.25409468
3	17	1	0.889	0.78967934	0.59104315	3.16275003
4	16	1	0.889	0.78945918	0.61216165	3.08000556
5	15	1	0.888	0.78802338	0.62904091	3.01223699
6	14	1	0.886	0.78558842	0.64264736	2.95647781
7	13	2	0.884	0.7807387	0.65117521	2.92098817
8	12	3	0.880	0.7752289	0.65795703	2.89245402
9	11	4	0.878	0.77174234	0.66712424	2.85342993
10	10	4	0.874	0.76393546	0.66950964	2.84318767
11	9	5	0.868	0.75329098	0.6678917	2.85013866
12	8	4	0.864	0.74662941	0.67155664	2.83436886
13	7	3	0.856	0.73265115	0.66581394	2.85904041
14	6	2	0.840	0.70632243	0.64556155	2.94439819
15	5	4	0.825	0.6809907	0.62782248	3.01717983

# Var: number of variables in the initial saturated model, # sig var: number of significant variables, Sig: significant. Adjusted for: does not meet physical activity recommendations, does not meet 30 min per day of exercise (at least 5 days), does not meet recommendations for vigorous physical activity, primary incomplete (0–4 years), first quartile of wealth, origin: urban or rural, BMI risk classification, consumption food pattern 1 (traditional), consumption dietary pattern 2 (industrialized), consumption dietary pattern 3 (preservative), consumption dietary pattern 4 (food drinks-grill, special diet, no vegetarian diet, subsidized affiliation regime/not affiliated, have high blood pressure, have diabetes). * R^2^ avoids overestimation of goodness of fit.

**Table 5 nutrients-15-04889-t005:** Model Factors of greatest influence on the incidence of cervical cancer.

Variable	Bivariate		Multivariate	
	R^2^ (%)	*p*-Value	Standardized β	*p*-Value
Village/Rural	13.1	0.0300	−0.40	0.003
Conservative pattern consumption	18.5	0.0089	−0.28	0.013
Subsidized/Unaffiliated	18.1	0.0096	0.43	0.002
Have Diabetes	30.8	0.0004	0.37	0.007

Adjusted for: subsidized/unaffiliated affiliation, housing area town or rural center, consumption dietary pattern 3 (conservative), vegetarian diet, have diabetes.

## Data Availability

The data will be made available to anyone who requests it from the corresponding author through a reasoned request.

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
