# Peer review of "Association between Cervical Cancer and Dietary Patterns in Colombia"

_nutrients, 2023, doi:10.3390/nu15234889_

Round 1
Reviewer 1 Report
Comments and Suggestions for Authors
To The Chief Editor
Journal of Nutrients
The manuscript “Association between cervical cancer and dietary pattern in Columbia” is well written and a very good attempt to evaluate how dietary intake is associated with risk of cervical cancer. Before this paper can be accepted for publication some points need to be addressed.
Major issues.
1. As the title suggests, there is an association between dietary intake and cervical cancer, but Author did not provide the list of dietary intakes that promote or prevent from Cervical Cancer.
2. Information was collected from 3472 women belonging to the population aged 35-64 years, but Author did provide any medical details of these patients whether these patients have any other disease.
3. Data in the result is mostly validated by ecological model but according to the title it should be validated more by research findings based on dietary intake and it should be more details on the mechanism of how dietary intake can improve the prognosis of cervical cancer.
Comments on the Quality of English Language
Needs to be improved
Author Response
Dear Editor and reviewer:
First, we would like to thank the Editor and reviewer for the time dedicated to revising the manuscript, their assessments, and all the comments made on this work. We agree with most of the observations and are convinced that they have improved the clarity and scientific value of this research.
All changes in the revised manuscript have been pointed using yellow color.
We attached the modified version via the web platform.
The authors.

Reviewer 2 Report
Comments and Suggestions for Authors
In this study, the authors investigate the correlation between dietary practices and the occurrence of cervical cancer in a sample of 3,472 Colombian women aged between 35 and 64. Drawing from two national datasets, the research emphasizes the protective attributes of conservative dietary choices and rural residency in counteracting cervical cancer. It is worth noting the inclusion of other factors, such as state-sponsored health program affiliation and diabetes mellitus, as key determinants. These findings advocate for enhanced policy measures and public awareness campaigns to address the risk of cervical cancer. While the paper is an informative contribution to ongoing cervical cancer prevention efforts globally, it requires some revisions and elucidations for potential publication. My specific comments are:
1. Can the authors provide data indicating whether the studied women have given birth or not?
2. Table 2: In the "overweight classification" row, certain numbers appear as decimals rather than percentages. The same issue is present in the “obesity classification” row.
3. Figures 1 and 2: can the authors label the specific regions on each map?
4. Table 4: A clear description of the methodology used to adjust the R-squared value is necessary.
Author Response

(The authors gave the same response as above.)
